# Gluing Techniques on Bond Performance and Mechanical Properties of Cross-Laminated Timber (CLT) Made from *Larix kaempferi*

**DOI:** 10.3390/polym13050733

**Published:** 2021-02-27

**Authors:** Mingyue Li, Shuangbao Zhang, Yingchun Gong, Zhaopeng Tian, Haiqing Ren

**Affiliations:** 1Research Institute of Wood Industry, Chinese Academy of Forestry, Beijing 100091, China; lmy125@foxmail.com (M.L.); yingchungong@caf.ac.cn (Y.G.); tianzhaopeng@caf.ac.cn (Z.T.); 2College of Materials Science and Technology, Beijing Forestry University, Beijing 100083, China; shuangbaozhang@bjfu.edu.cn

**Keywords:** cross-laminated timber, *Larix kaempferi*, bond performance, mechanical properties, gluing technics

## Abstract

Previous studies have proved that *Larix kaempferi* is a good material for preparing cross-laminated timber (CLT), but under bending shear stress, CLT made by *Larix kaempferi* is prone to the phenomenon of bonding face cracking, which seriously affects the shear performance of CLT. To solve this problem, this paper took *Larix kaempferi* as raw material, conducted experiments on the surface sanding conditions, gluing pressure and adhesive types of sawing timber, and explored the influence of these three factors on the bonding quality of CLT. The microscopic characteristics of the bonding layer were further studied. The results showed that for *Larix kaempferi* with a density of 0.68 g/cm^3^ used in this experiment, a high bonding pressure is required. Among the three cold curing adhesives selected in the experiment, emulsion polymer isocyanate (EPI) adhesive needs 1.5 MPa bonding pressure to ensure the bonding quality, while for polyurethane (PUR) and phenol resorcinol formaldehyde (PRF), 1.2 MPa can meet the need of adhesive pressure. This is concerned with the permeability of different adhesives under different pressures. The microscopic results of the bonding layer show that EPI adhesives have poor permeability, so it requires high bonding pressure. The influence of sanding surface of different sand-belt on block shear strength (BSS) and wood failure percentage (WFP) is not obvious, while the durability of bonding layer is better when sanding mesh number is 100. Hence, a high pressure should be used for CLT industrial production when the laminate density is higher, especially when the adhesive has poor permeability. Reasonable sanding surface treatment can be used in laminate surface treatment to improve the durability of CLT.

## 1. Introduction

Cross-laminated timber (CLT) is a kind of timber structural building material made of wood or other sawn timber, which is connected with each other in a vertical direction with adhesive. Due to the unique structure of CLT, its format can be extended indefinitely and it has good mechanical properties. Wood construction has become increasingly popular due to various sustainability advantages, hence, CLT can be used as the material of tall timber structure buildings particularly in congested urban centres such as North America, Austria, Germany, Australia, et al. [1]. As a country with large population, tall wooden buildings are urgently needed in China, and CLT is becoming the preferred material for tall wooden buildings due to its unique properties. The production of CLT with domestic wood can reduce the transportation cost, and the production of CLT with domestic wood has gradually become a hotpot.

Timber-terrazzo composite floor and other inorganic material which have good mechanical behaviors are often used in traditional buildings [2]. Wood structure has been recognized worldwide due to its environmental protection and sustainable growth characteristics. CLT originated in Europe, but *Larix Kaempferi* was rarely used to prepare CLT in Europe. In China, *Larix kaempferi* is the main afforestation and timber species in the subalpine region, which has good mechanical properties and suitability for wood products. The density of *Larix kaempferi* ranges from 0.48 g/cm^3^ to 0.84 g/cm^3^ depending on location and environment. The rolling shear modulus of CLT prepared by *Larix kaempferi* was about 140 MPa [3], which was much higher than that prepared by *Picea sitchensis*, *Abies fabri*, and other softwood in North America [4,5]. Previous studies have shown that the bonding property is one of the important properties for the adhesive products made from *Larix kaempferi* [6,7]. Compared with the softwood species commonly used for CLT in Europe, hardwood and softwood with higher density are more difficult to have good bonding performance due to its high density, and resin content [8,9], and the same problem exists with CLT. Therefore, it is of great industrial interest to explore effective techniques for enhancing the bond performance of *Larix kaempferi* CLT.

There are several types of adhesives that have been used for structural adhesives including phenol resorcinol formaldehyde (PRF), polyurethane (PUR), melamine urea formaldehyde (MUF), and emulsion polymer isocyanate (EPI). The selection of a suitable adhesive depends very much on the species of wood. The integrity of a wood-adhesive bond is of particular importance to the end use of glued products. Bonding performance is significantly influenced by adhesive species, species of wood and surface treatment. Yusof et al. [10] prepared CLT by using *Acacia mangium* as raw material with different adhesives. The results showed, that the durability of PRF glued CLT is better than that of PUR. Moreover, with the increase of bonding pressure, the shear strength of CLT increases significantly, so for *Acacia mangium* a higher bonding pressure is required. PRF has better bonding performance than PUR, which is because PRF has better permeability for *Acacia mangium*, and the permeability of adhesive is an important factor affecting the bonding performance [11]. Previous studies have shown that different adhesives and bonding technics are required for different wood species, and studies on the preparation of CLT of *Larix kaempferi* are worth further study.

Sanding is a widely-used method to create smooth, homogeneous surfaces before gluing [12]. Test study indicates that bonds of surfaces sanded with coarse grit (36) performed poorly, with finer grit (80–180) resulted in good performance [13,14]. While sanded wood surfaces showed crushed and torn-out fibrils [14,15]. Partially detached or slightly crushed cell wall components named fibrillation, which is considered to contribute to a good bonding quality and the wood surface sanding can effectively improve the durability of the adhesive layer [16]. However, it has been given little attention as a pretreatment method for bonding of CLT.

Hence, the objective of this study was to (1) select an efficient bonding method for CLT made from *Larix kaempferi*, (2) find a suitable surface treatment method to prepare CLT.

## 2. Materials and Methods

### 2.1. Material Preparation

*Larix kaempferi* used in this study is from the artificial forest growing in Liaoning province, China. The average oven-dried density was 0.68 g/cm^3^. All wood logs were sawed to 21 mm (Radial) × 90 mm (Tangential) × 540 mm (Longitudinal). According to GB/T 29897-2013 [17], visually classify all the timbers should be carried out before making CLT, only those that met the requirements of Grade No. 2 in the North American lumber grading system will be kept for further processing. In order to make the moisture content of the wood around 12%, the timber should be kept under the condition of 25 °C and 65% relative humidity at least four weeks. The material with obvious defects is discarded, the remaining wood is graded for density, and the less dense wood is used as the intermediate layer.

Three commercial cold curing, an EPI (Harbin, China), PRF (Shenyang, China), and PUR (Harbin, China), were used for manufacturing CLT.

### 2.2. Surface Sanding Pretreatment

To improve the bonding performance of *Larix kaempferi*, the surface of the sawn timber were sanded with different sand belts (P60, P80, and P100), sanding feed speed (v_f_) is 6 m/min, grinding depth (α_p_) was set as 0.3 mm.

### 2.3. Surface Microscopy

The surface textures were observed by SEM, small specimens (10 mm × 10 mm × 5 mm) were prepared and coated with a thin layer of gold. Micrographs were taken at 10 KV acceleration voltage using a Hitachi “S-4800” microscope with Bruker Skvscan “1172” system (Tokyo, Japan).

### 2.4. Surface Roughness

The surfaces’ roughness were measured with Mitutoyo profilometer (Shanghai, China). The stylus tip was 90°cone angle and tip radius is 5 μm. The measurements speed is 0.25 mm/s. Four roughness parameters R_a_ (arithmetical mean deviation), R_z_ (The distance between the contour peak line and the contour valley line) and R_t_ (The sum of the average height of the peak of the maximum contour and the average depth of the valley of the maximum contour), and R_sm_ (The average spacing of the microscopic irregularities of the contour) were calculated.

### 2.5. CLT Manufacturing

Three different adhesives (EPI, PRF, and PUR) and three levels of bonding pressures (0.8 MPa, 1.2 MPa, and 1.5 MPa) were used in this study and the lumber’s size were detailed in 2.1. The manufacturer’s preparation instruction of each adhesive was shown in Table 1. The method of gluing is manual gluing, and the pressure provided by the press (Figure 1).

To study the effects of surface sanding treatment on bonding performance of *Larix kaempferi* CLT, sanding belts with P60, P80, and P100 were used to change the surface roughness of the laminate. Gluing conditions are shown in Table 2.

### 2.6. Test of Permeability of Adhesive

In order to observe the permeability of the adhesive under different bonding pressures, 8–15 μm thick slices including bonding line were cut from D in Figure 2 and dyed. The permeability of bonding layer was observed under fluorescence microscope and optical microscope. Six repetitions are conducted for each gluing technics.

### 2.7. Block Shear Tests

CLT blocks (B) were cut from the geometric center of CLT panel as showed in sampling diagram according to ASTM D2559 [17]. Block shear tests were conducted according to [18,19] and Ehrhart’s [20]. Wood failure percentage (WFP) and block shear strength (BSS) were determined according to ASTM D905 [19] and D5266 [21], respectively. 10 repeats for each conditions.

### 2.8. Cyclic Delamination Tests

Cyclic delamination (vacuum-pressure-soak rapid dying) tests were conducted to obtain the rate of delamination (RD) according to AITC Test T110-2007 [22]. The sampling location is C, as shown in Figure 2. 10 samples were tested for each gluing technics.

### 2.9. Short Span Center-Point Bending Tests

The interlaminar shear strength of *Larix kaempferi* CLT (580 mm × 90 mm × 63 mm) was determined via short span center-point bending tests according to ASTM D 3737 [23]. Tests were conducted at a span-to-ratio of 6.5. The sampling location is A, as shown in Figure 2. Six replicates were tested for major strength directions.

## 3. Results and Discussion

### 3.1. Effects of Adhesives and Pressures on Bonding Performance of Larix kaempferi CLT

#### 3.1.1. Block Shear Performance

Effects of different adhesives and pressure on CLT block shear strength and percentage of wood failure are presented in Table 3. All three adhesives in the tests had the best adhesive performance under high bonding pressure (1.5 MPa), while the BSS average value was lower under low bonding pressure and the degree of dispersion was higher. The WFP of samples under 1.5 MPa pressure was the highest, and the WFP of all the 10 samples in each group was above 75%. When bonding pressure were 1.2 MPa and 0.8 MPa, the WFP of some samples manufactured with EPI and PUR could not reach 75%.

When bonding pressure were changed, the BSS average values of CLT bonding with PUR were pretty much the same. While the WFP was increases as the pressure increases. The BSS average values of CLT bonding with EPI and PRF were obviously increased, and the variation regularity of WFP was the same with that bonded with PUR. When the bonding pressure was 1.2 MPa and 1.5 MPa, BSS average value and its dispersion degree of PRF showed no significant change, and the WFP of all the 10 samples was above 75%. It can be concluded that to prepare *Larix kaempferi* CLT bonded with PUR and PRF, 1.2 MPa bonding pressure can be used, while bonded with EPI a higher bonding pressure of 1.5 MPa is required. Under the condition of pressure change, the change of the coefficient of variation (COV) value is also one of the important indexes to evaluate the bonding quality. When the COV value is smaller, the data is more stable. However, the influence of increasing pressure on the numerical stability of BBS is not obvious.

#### 3.1.2. Cyclic Delamination Performance

Effects of adhesives and bonding pressures on the cyclic delamination performance of *Larix kaempferi* CLT are shown in Table 3. With the increase of bonding pressure, the RD value of all the three adhesives decreased. The durability of EPI adhesive was more susceptible to bonding pressure than the other two, when the adhesive pressure was 1.5 MPa the RD value decreased by 43.5% compared with 0.8 MPa bonding pressure. Among the three adhesives selected in this study, the durability of PUR is less affected by adhesive pressure. It can be seen from the COV that when the pressure increases, it not only reduces the delamination rate but also improves the stability of durability.

#### 3.1.3. Interlaminar Shear Strength

Interlaminar shear strength and displacement of *Larix kaempferi* CLT tested by short span center-point bending are shown in Figure 3. The average interlaminar shear strength of CLT prepared by *Larix kaempferi* under different conditions is 3.16 MPa. It was reported that the interlaminar shear strength in the major strength direction for three-layer poplar CLT with a thickness of 105 mm was 2.0 MPa, for three-layer black spruce CLT with a thickness of 105 mm was 1.6 MPa, for three-layer Eucalyptus CLT with a thickness of 54 mm was 1.75 MPa [8,24,25].

When the bonding pressure was 0.8 MPa, the values of interlaminar shear strength of the three adhesives (PUR, EPI, PRF) were 3.37 MPa, 2.29 MPa, and 3.33 MPa, at the meanwhile, the displacement of center of span were 7.07 mm, 4.91 mm, and 5.69 mm, respectively. When the adhesive pressure was 1.2 MPa, the values of interlaminar shear strength of the three adhesives (PUR, EPI, PRF) were 3.05 MPa, 2.78 MPa, and 2.92 MPa, and the displacement of center of span were 7.17 mm, 5.12 mm, and 5.61 mm, respectively. When the adhesive pressure was 1.5 MPa, the values of interlaminar shear strength of the three adhesives (PUR, EPI, PRF) were 3.52 MPa, 3.39 MPa, and 3.86 MPa, while the displacement of center of span was 7.96 mm, 7.00 mm, and 6.92 mm. The data analysis showed that the results are the same as those of BSS. For the three adhesives used in the test, the pressure of 1.5 MPa has higher interlaminar shear strength and lower dispersion degree.

#### 3.1.4. Microscopic Characterization of the Bonding Interface

It is considered to be an effective method to study the gluing property deeply by detecting the microscopic characteristics of the bonding interface [26]. The influence on bonding interface layer of three adhesives under three different bonding pressures were observed under laser confocal microscope and optical microscope. As shown in Figure 4, PUR adhesive is a typical foaming adhesive. When the bonding pressures were 1.2 MPa and 1.5 MPa, it can be observed that the PUR adhesive penetrates into wood cells, especially when the pressure is 1.5 MPa, there are more adhesives that can be seen in wood cells, while when the pressure is 0.8 MPa, there is no adhesive that can be observed in wood cells. For EPI adhesive, the adhesive penetrated into the wood cells when the bonding pressure was 1.5 MPa, but when the gluing pressures were 0.8 MPa and 1.2 MPa, there are no adhesive can be seen in wood. PRF adhesive has good permeability, and it can be observed to penetrate into wood at three different gluing pressures. Different gluing pressure should be selected according to the permeability of adhesives. For *Larix kaempferi* with high density selected in this experiment, higher gluing pressure is more suitable.

### 3.2. Effects of Surface Roughness on Bonding Performance of Larix kaempferi CLT

#### 3.2.1. Surface Roughness of Larix kaempferi after Sanding with Different Sand Belt

The comparison of different mesh number sanded surfaces revealed differences for all roughness values. *Larix kaempferi* is softwood with a uniform structure. The surface roughness of *Larix kaempferi* treated by different sand belts is different. The results of the surface roughness measurements are shown in Table 4. With the sanding belt mesh number increased, the parameters of surface roughness is reduced. In addition, Figure 5 displays the differences between the three surface treatment methods. Sand-belt treated wood surface with mesh 80 and mesh100 produced a higher level of fibrillation.

#### 3.2.2. Block Shear Performance

The results for BSS (mean and coefficient of variation (COV)) and WFP together with statistical analyses are shown in Table 4. The BSS values of the three sand belts were not obviously different, and the WFP was above 75%, which had better adhesive quality. Sanded surfaces showed higher value of R_a_ and R_z_, and associated with this, a larger surface is available for adhesion as Markus’s research [16], while larger surface are not directly related to BSS and WFP in this study. The results did not show a clear correlation between roughness and BSS or WFP, which were in agreement with findings from Kläusler et al. [13].

#### 3.2.3. Cyclic Delamination Performance

The RD values have been evaluated with regard to standard requirements [22], with the increase of the mesh number of sand belt, the surface roughness decreases and the RD value decreases gradually. When the sand-belt mesh number is 60, 80, and 100, the RD is 12.8%, 9.02%, and 5.43%, respectively. When the number of sand-belt mesh is 100, more fibrillation is generated to enhance the cyclic delamination performance of the bonding interface, which is the same as previous study [16]. Therefore, for *Larix kaempferi*, the durability of CLT can be improved by sanding the surface of sawn timber with 100 mesh sanding belts.

## 4. Conclusions

Effects of adhesives and pressures on bonding performance and mechanical properties of CLT manufactured by *Larix kaempferi* were explored, a nd the sanding treatment to prepare laminates was discussed. The main findings are summarized as follows:1. Bonding pressure is determined by the permeability of the adhesive and the wood, which is also the main factor affecting the bonding performance, especially for EPI adhesive. It is recommended to use high bonding pressure to control the consistency of product quality for industrial purposes.2. The surface of the treated laminate with different mesh number of sand belts has no obvious influence on the BBS, but has a great influence on the cyclic delamination performance. The cyclic delamination performance is good when the sand belt mesh number is 100. Therefore, suitable sanding pretreatment can be one of the methods to improve the durability of CLT bonding line.

## Figures and Tables

**Figure 1 polymers-13-00733-f001:**
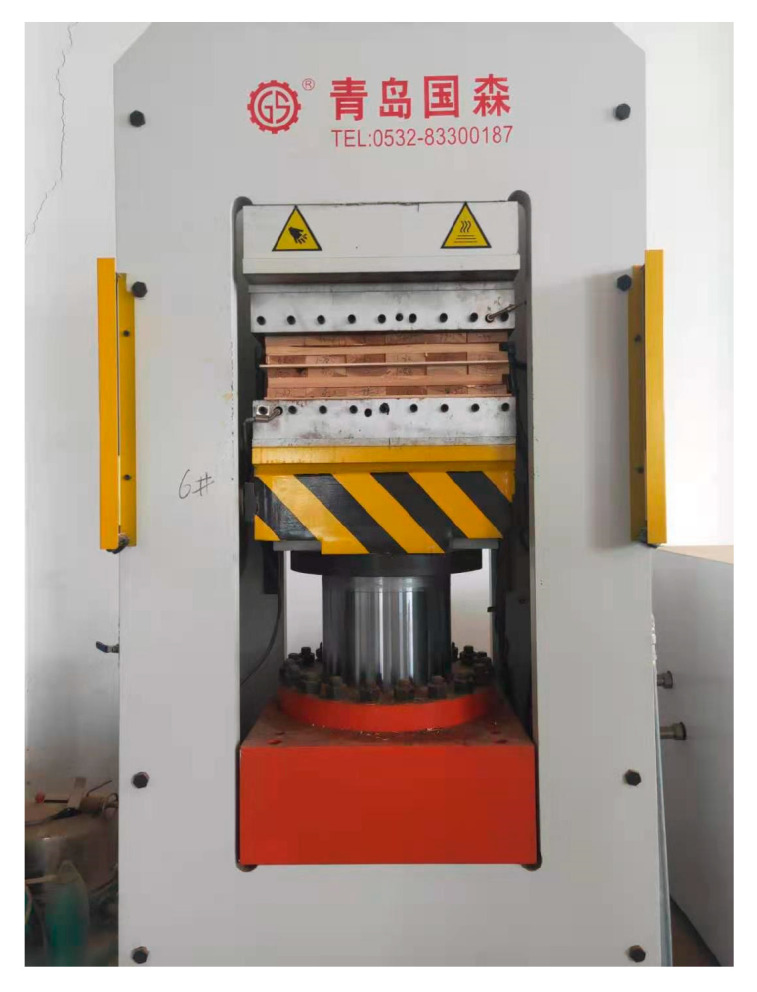
Image of cross-laminated timber (CLT) manufactured by press in the lab.

**Figure 2 polymers-13-00733-f002:**
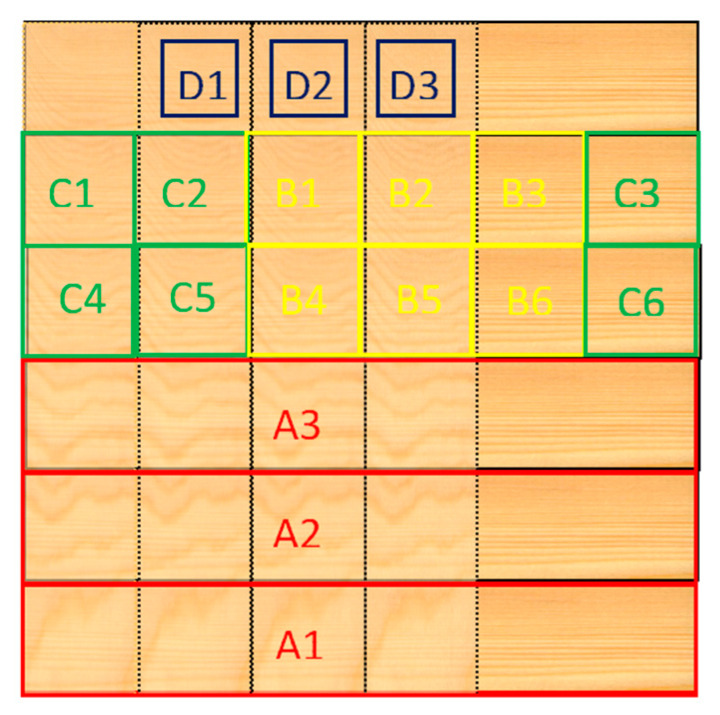
Sampling diagram of CLT.

**Figure 3 polymers-13-00733-f003:**
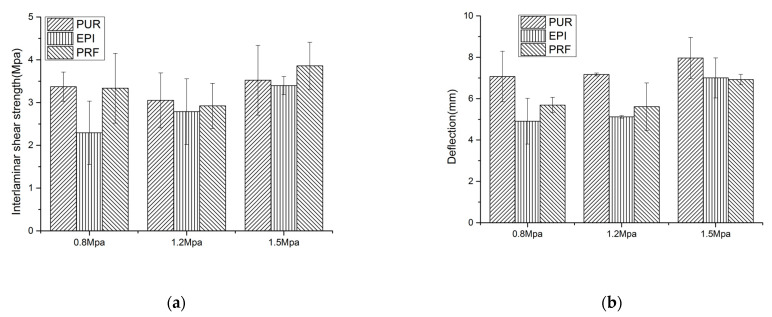
Mechanical properties of CLT under short span center-point bending test(**a**. interlaminar shear strength; **b**. Deflection).

**Figure 4 polymers-13-00733-f004:**
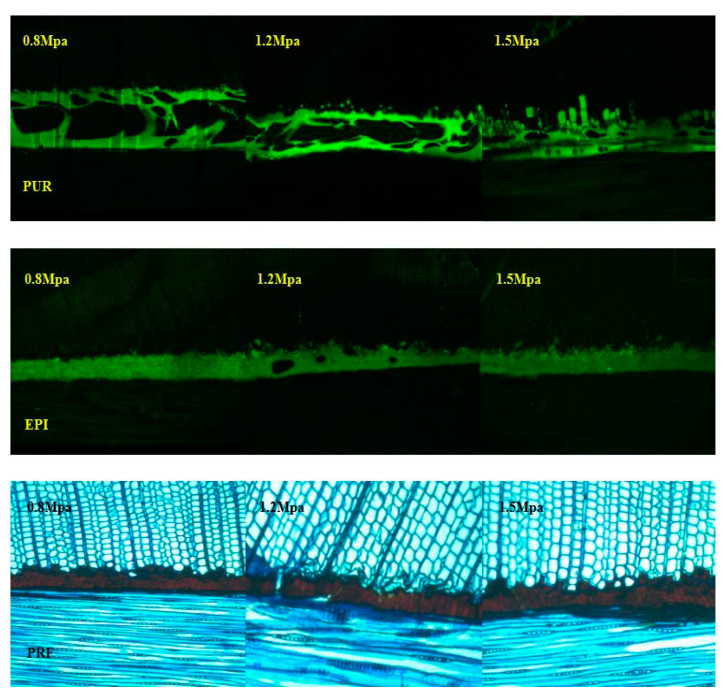
Fluorescence and optical microscope photomicrographs with transverse view of horizontal bond lines showing the penetration of adhesives.

**Figure 5 polymers-13-00733-f005:**
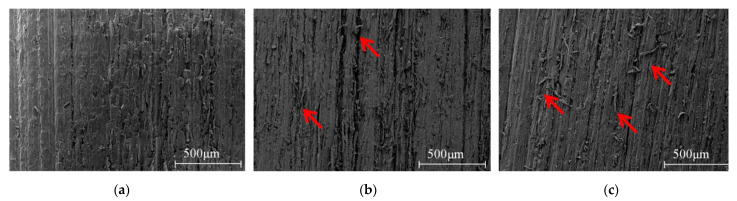
SEM of tangential sections, which were sanded with different mesh sanding belts. (**a**) mesh 60, (**b**) mesh 80, (**c**) mesh 100; the red arrows marked the fibrillation.

**Table 1 polymers-13-00733-t001:** Gluing process parameters.

Adhesives	Spread Rate (g/m^2^)	Assembly Time (min)	Pressing Time (min)	Curing Temperature (℃)
EPI	300–320	30	160	30
PRF	380–400	50	180	30
PUR	180–200	70	200	30

**Table 2 polymers-13-00733-t002:** Sanding conditions, adhesives, and pressure.

Abrasive Belt Mesh	Adhesive	Pressure	Temperature
P60/P80/P100	PUR	1.5 MPa	30 ℃

**Table 3 polymers-13-00733-t003:** Effects of adhesives and bonding pressures on wood failure percentage (WFP), block shear strength (BSS), and rate of delamination (RD) (COV means coefficient of variation).

Pressure	Adhesives	BSS (COV)	WFP (Number < 75% WFP)	RD (COV)
0.8 MPa	PUR	1.99 MPa (11.44%)	83.5% (2)	17.85% (27.73%)
0.8 MPa	EPI	1.59 MPa (2.27%)	45% (5)	55.6% (55.10%)
0.8 MPa	PRF	1.52 MPa (3.6%)	85% (3)	29.7% (41.35%)
1.2 MPa	PUR	1.98 MPa (13.25%)	70% (2)	15.3% (9.42%)
1.2 MPa	EPI	1.53 MPa (44.7%)	37.5% (6)	43.9% (4.66%)
1.2 MPa	PRF	2.11 MPa (17.0%)	77.5% (0)	17.2% (10.36%)
1.5 MPa	PUR	2.02 MPa (3.3%)	97.5% (0)	8.35% (5.32%)
1.5 MPa	EPI	2.03 MPa (25.4%)	94% (0)	12.1% (1.58%)
1.5 MPa	PRF	1.96 MPa (20.2%)	96% (0)	9.01% (2.02%)

**Table 4 polymers-13-00733-t004:** Surface roughness of specimens prepared with different sanding belts and its effects on BSS and RD.

Mesh Number of Sand Belts	R_sm_ (mm)	R_a_ (μm)	R_z_ (μm)	R_t_ (μm)	BSS (COV)	WFP (N < 75%)	RD (COV)
P60	0.12	7.55	43.65	56.45	1.99 MPa (11.4%)	87% (0)	12.8% (3.43%)
P80	0.10	6.68	40.20	51.39	2.02 MPa (3.38%)	93% (0)	9.02% (2.81%)
P100	0.09	5.55	36.76	42.87	1.98 MPa (13%)	89% (0)	5.43% (3.56%)

## Data Availability

Data sharing not applicable.

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
