# Peer review of "Gluing Techniques on Bond Performance and Mechanical Properties of Cross-Laminated Timber (CLT) Made from Larix kaempferi"

_polymers, 2021, doi:10.3390/polym13050733_

Round 1

Reviewer 1 Report

Dear authors,

even if the paper is interesting, I feel the novelty of the research is not well described. Is Larix kaempferi conventionally used for CLT? Are the methods or adhesives you propose different from typical ones?

The number of specimens and tests is only given in section 2.9. How many CLT were manufactured for each adhesive and pressure tested? Why these levels of pressure? how many tests of permeability were perfomed? Same for delamination assays. 

Please, change "Mpa" in figures 1 and 2 to "MPa". 

Reviewer 2 Report

See the attached document, which is the review for the Authors.

Reviewer 3 Report

This paper studies the influence of surface sanding conditions, gluing pressure and adhesive types on the bonding quality of one specific (Larix kaempferi) cross-laminated timber (CLT).

Three cold adhesive types have been selected, named EPI, PRF and PUR, all fabricated and commercialized in China. The authors selected three bonding pressures, 0.8 MPa, 1.2 MPa and 1,5 MPa, and three different sand belts, P60, P80 and P100

The authors have found that for a good bonded performance of the CLT, high pressures should be used and that the sanding pretreatment would improve the durability of CLT bonding line. My personal opinion is that these results are obvious but the specific values of the variables (pressure, sand belt and kind of adhesive) can be useful for the timber construction industry-

However, some questions should be clarified by the authors:

  • There is confusion about the specimens. Wood logs were sawed to 21 mm (R) x 80 mm (T) x 540 mm (L), (Paragraph 2.1) What is T?, small specimens (10mm x 10mm x 5 mm) (Paragraph 2,3) Where these specimens come from? The lumber size were detailed in 2.1 (Paragraph 2.5) 8-15mm thick slices were cut and dyed (paragraph 2.6) From where they were cut?. 10 CLT blocks were cut from the geometric center of CLT panels (Paragraph 2.7). CLT (580 mm x 90 mm 63 mm) (paragraph 2.9). I can not follow the description of all these different specimens
  • I suppose that COV is the degree of dispersion, What is it? Standard deviation SD?
  • The second paragraph of point 3.1.1 is just to repeat the numbers on Table 3. It is completely unnecessary. The same can be said for the paragraph above figure 1, only a description of the values of figure 1
  • Table 3 is difficult to follow. The authors want to present the dependence of BSS and WPF on three different pressures and for three different adhesives. Perhaps, a graphical representation is clearer
  • I suppose that the studies of points 3.2,2 and 3.2.3 are under the conditions of Table 2. I suggest remembering it before describing table 5. Why is the reason to select PUR adhesive and 1,5 MPa?.

Round 2

Reviewer 1 Report

Changes have been performed, and the manuscript is now easier to follow trough.

Reviewer 2 Report

I recommend that the revised version of the article that has been resubmitted is accepted and published.

The Authors have done a good job, since they have considered how I had commented their article and have suitably and carefully addressed all my comments.

Moreover, in the revised version resubmitted, the appropriate structure and language have been used and the presentation is good and consistent, now. In particular, the description of the new methodology is accurate and clear.

Now, the article adds to the subject and the presentation saves the readers’ effort to understand the article.